# A Study of Aggregation of Long Time-series Input for LSTM Neural Networks

## Abstract

Time series forecasting is the process of using time series data to create a prediction model. Long-short term memory (LSTM) models are the state-of-the-art for time-series forecasting. However, LSTMs can handle limited length input mostly since when the samples enter the model in sequence, the oldest samples need to propagate through the LSTM cells self loop for each new sample and thus their data diminishes in this process. This limits the length of the history that can be used in the training for each time epoch. The common way of handling this problem is by partitioning time records to uniform intervals, averaging each interval, and feeding the LSTM with rather short sequences, but each represents data from a longer history.

In this paper, we show that this common data aggregation method is far from optimal. We generalize the method of partitioning the data, and suggest an Exponential partitioning. We show that non-uniformly partitioning, and especially Exponential partitioning improves LSTM accuracy, significantly. Using other aggregation functions (such as median or maximum) are shown to further improve the accuracy. Overall, using 7 public datasets we show an improvement in accuracy by 6% to 27%.

## 1 Introduction

Predicting the future has always been a task humans found hard to accomplish. In the last decade, many systems have surrounded themselves with multi-variate tracing, recording many variables throughout time. Natural systems such as weather and rivers, financial systems such as the stock market and Bitcoin and global epidemics, all are recorded.

This rise of big data allowed machine learning algorithms to predict the future with great accuracy. The mentioned systems can be described using a single variable over time, namely univariate time series, or several variables altogether as a multivariate time series. In both cases, one variable is marked as the important one whom we'd like to predict.

However, typically, machine learning models are not capable of handling long time series. As the input intake to the model (more samples) grows wider, the model is required to adjust more internal weights to learn the dependencies in the data. So while additional data appears to be always an advantage, it affects learning speed but also learning accuracy.

To overcome the trade-off between the will to accommodate as much data as possible and the engineering limitation of the learning process it is common to aggregate the data at the model input. Namely, a long data sequence is used, but the input to the model is kept small by partitioning the data to equal size intervals and using the average of the samples in each interval (Li et al., 2017; Yu et al., 2017; Petersen et al., 2019; Zheng et al., 2020; Farhi et al., 2021). This enables to preserve the model size modest while still looking far to the past.

We observe here that this standard way of aggregation cannot be optimal since obviously the data closer to the present ("newer") is more important than older data, but it is still aggregated similarly. Thus, we suggest an aggregation method that increases the aggregated interval length exponentially as we look farther back in time. Additionally, we question the use of averaging as the best aggregation function.

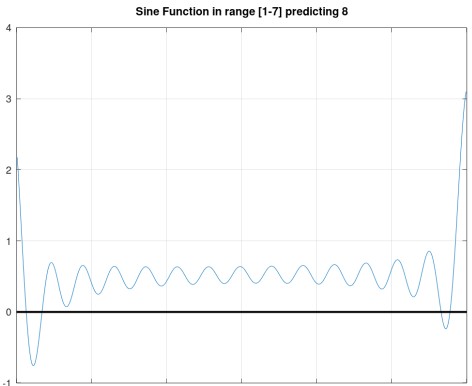 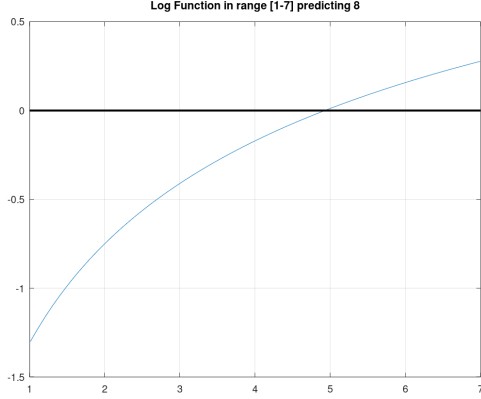

Figure 1: The stimation error for optimal Linear estimators as a function of the interval partition point.

In this paper, we study our *Exponential partitioning* and show using seven publicly available datasets that indeed it performs much better than the uniform partitioning that is commonly practiced – an improvement of 6.4-27% in the RMSE of the prediction. Interestingly, the best results were usually achieved for slow Exponential grows where the exponent base is in the range $[1.05, 1.1]$.

We also study other aggregation methods for the intervals (sample partitions) and found a more complex picture. Median was the best aggregation function for most of the datasets, however average was still the best for others, and in one case Maximum was the best aggregation function. We also tested a combination of two functions but could only achieve minor improvements. The source-code for our Exponential partitioning is available at `http://www.github.com/TBD`.

## 2 MOTIVATION

To motivate our approach, consider the simplest case of estimating the future value of a number series using a linear estimator, which is based only on two values of the series. To model this, suppose the series is generated by some function and we can sample the function in the interval $[x_s, x_e]$ and want to predict the value at $x_p > x_e$.

Similarly to the practice in long time series entry to LSTM, instead of selecting the series (function) value at two points in the interval, we can calculate the average of sub-intervals. Equation 1 and Equation 2 show the calculated two points, as a function of $k$, a value within the interval (not necessarily the middle of it).

Equation 3 is the linear estimator and Equation 4 calculates the difference between the actual value and the predicted value using a linear estimator. Figure 1 show the calculated differences for the Sine $(sin(15 \cdot X))$ and Log $(\log_2 X)$ functions with the linear interpolation that were done with the interval $[1, 7]$ where $k$ varies in the interval $(1, 7)$ and the predicted value is at 8. The graphs clearly show that the optimal partition is not in the middle of the interval and selecting a partition close to the end of the interval is optimal. Similar results were obtained with a second order Lagrange polynomial interpolation.

$$x_1(k) = \frac{k + x_s}{2} \quad x_2(k) = \frac{x_e + k}{2} \quad (1)$$

$$y_1(k) = \frac{\int_{x_s}^{k} f(x)\, dx}{k - x_s} \quad y_2(k) = \frac{\int_{k}^{x_e} f(x)\, dx}{x_e - k} \quad (2)$$

$$P(x, k) = x \cdot \frac{y2(k) - y1(k)}{x2(k) - x1(k)} \quad (3)$$

$$Diff(k) = F(x_p) - P(x_p, k) \quad (4)$$

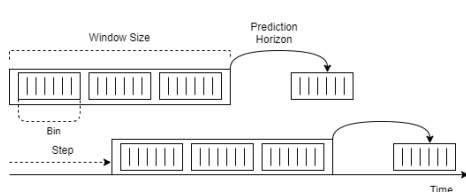
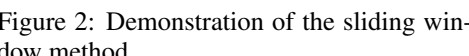
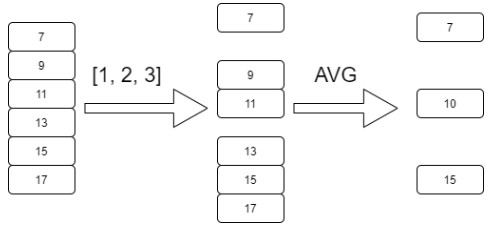

Figure 2: Demonstration of the sliding window method.

Figure 3: An example of non-uniform partitioning of time series data.

## 3 RELATED WORK

Different architectures implemented an ensemble model that tries to focus both on short term data and long term, for better accuracy. Krstanovic & Paulheim (2018) constructed an ensemble of LSTM models, each sub-model selected a different window size and other various parameters. Zhao et al. (2020) also created a cluster of LSTMs that are constructed to separately forecast with different time lag. NNCT weight integration strategy is introduced to determine the weight coefficients of the ensemble model. A more general method was proposed by Choi & Lee (2018) where an ensemble of LSTM models with varying sequence length (accordingly to different datasets), this allows the model to be capable of handling the dynamic different real-world time series. These methods, although trying to combine between different intervals of time, indirectly try to solve the problem, and do not create a single model that predicts accurately.

Changes to the sliding window algorithm were suggested by Laguna et al. (2011) who proposed a dynamic sliding window, where the size varies and changes according to the last selected window, and after is processed by Dynamic Bayesian Network. It must be noted that this type of pre-processing does not fit LSTM models since the input data shape is not static. Other sliding window methods, mostly for optimization purposes were suggested by Traub et al. (2019); Tangwongsan et al. (2017).

Similar mechanisms from the field of image vision have been imported into time-series analysis as an attempt to handle temporal data with many features (such as images) (Donahue et al., 2015). A single model, composed of CNN and LSTM, is used to handle a time-series input, where each instance in time contains many features. In this type of model, time series data is pre-processed by CNN to extract a fixed meaningful vector of the input. We note that unlike this type of model which deals with short window size and many features, this paper deals with long window size.

## 4 METHODS

In this section we describe how the input is processed for the LSTM model. Data is supplied as a list of time records, where for each time record multiple variables may be available. The data is fed in multiple cycles, where in each cycle only data from a specific time window is used.

### 4.1 LSTM INPUT

The LSTM architecture accepts its data as a series of timestamps, each can be associated with many features. The following process is employed to allow a time series to be given as input to LSTM. For a single input, we select a window in time and consider the data in this window to predict value of our predicted variable in a constant distance in the future, also called Prediction Horizon (PH). For example, an 8-hour window can be used to predict value that is a 4 hours ahead of the end of that window (Figure 2). We also note that the figure show a usage of one feature only, when many features could be used in parallel.

A relevant window size may contain hundreds of samples, and is too large for an LSTM. To lower the input size, many papers suggest aggregating the windows into larger time interval, each is represented by the average value of all samples in the interval. For example, if the window size is 3 hours and the interval size is 1 hour, a single input to the LSTM is 3 vectors each is an average of 1 hour. The next input to the LSTM is generated by sliding the window by a certain time period, termed a step (Figure 2). The prediction result can be either a regression, namely an attempt to predict the

| Index | partition | $\varepsilon$ | base ($b$) |
|---|---|---|---|
| 0 – uniform baseline | {6, 6, 6, 6, 6, 6, 6, 6} | – | – |
| 1 – simple baseline | {1, 1, 1, 1, 1, 1, 1, 1} | – | – |
| 2 | {1, 1, 1, 1, 1, 1, 1, 41} | – | – |
| 3 | {1, 2, 3, 4, 5, 6, 7, 20} | – | – |
| 4 | {4, 4, 4, 4, 4, 4, 4, 20} | – | – |
| 5 | {1, 1, 1, 2, 2, 2, 10, 29} | – | – |
| 6 | {1, 1, 1, 2, 4, 8, 15, 16} | – | – |
| 7 | {1, 1, 1, 2, 2, 3, 3, 35} | 0.25 | 1 |
| 8 | {1, 1, 2, 2, 3, 3, 4, 32} | 0.30 | 1 |
| 9 | {2, 2, 2, 2, 3, 3, 3, 31} | 0.10 | 2 |
| 10 | {1, 1, 1, 1, 1, 2, 2, 39} | 0.15 | 1 |
| 11 | {1, 1, 1, 2, 2, 2, 2, 37} | 0.20 | 1 |
| 12 | {1, 1, 1, 2, 2, 2, 3, 36} | 0.23 | 1 |

Table 1: Selected partitions for a window of 48 samples, with the parameters for Equation 6.

value, or it can be a classification problem such as predicting that a value will rise above a certain threshold.

## 4.2 WINDOW PARTITIONING

We suggest to model aggregation as a partition of an interval (time window), as defined in Equation 5, where $x_0$ and $x_n$ are the boundaries of the window, $x_n - x_0$ is the window size, and $[x_i, x_{i+1}]$ is a sub-interval of the partition.

$$x_0 < x_1 < x_2 < ... < x_n \tag{5}$$

Since each subinterval in time holds a countable number of samples, we will also use the term *bin* for a subinterval, and use the term bin size for the number of samples in the sub-interval. We will refer to the partition by the sequence of bin sizes; for example, if the window holds 12 samples, {3,4,5} is a partition into three subintervals and the number of samples in the three subintervals is 3, 4, and 5.

The common practice (Farhi et al., 2021; Li et al., 2017; Yu et al., 2017; Petersen et al., 2019; Zheng et al., 2020) is to use a uniform partitioning, namely $x_{i+1} - x_i = x_{j+1} - x_j, \forall i, j$. We suggest here to use a non-uniform partitioning.

## 4.3 EXPONENTIAL PARTITIONING

As was explained in 2, we observe that the data with higher importance is the closer to the present. Thus, we suggest the Exponential partitioning where the bin size grows as we use data deeper in the past. Formally Equation 6 defines the bin sizes of an exponential partitioning. This is done until the penultimate bin, where the last bin holds the reminder of the window. Figure 3, shows an example of partitioning process, where the window size is 6 records and a list of bins is {1, 2, 3}.

Table 1 shows some of the partitioning we used in the paper and their exponential parameters when applicable. Partition number 0 is the common uniform partition (uniform baseline) and partition number 1 is the case where the last $n$ samples are taken as singletons and there is no aggregation at all (simple baseline) – both are used as baselines on our experiments. Partition number 2 takes the $n - 1$ last samples as singletons and aggregates the rest in one bin. This partition is important to understand whether the rest of the history is meaningful where compared to no aggregation (partition number 1 ) We show in Section 5.2 the impact of aggregating the window size's reminder is significant. Partition number 3 is a linear partitioning (bin sizes grow linearly), and it is closely resemble an Exponential partitioning with $b = 1$ and $\varepsilon = 0.45$

$$x_{i+1} - x_i = \lfloor b(1 + \varepsilon)^i \rfloor \tag{6}$$

### 4.4 AGGREGATION FUNCTIONS

The most common way of aggregating data in a bin is calculating the average of the samples. This is done as an attempt to reduce the dataset size while preserving the remaining data meaningful. Averaging is not necessarily the best way to do so, other functions such as minimum, maximum and median, may be proven more helpful. Indeed, we show that in many scenarios other aggregation functions represent actual data better than average. Other possible methods are calculating the median, sum or any other OWA operator (Yager, 1988; Rivero-Hernández et al., 2021).

### 4.5 DATASETS

We used a collection of 7 time-series datasets, where records are at constant intervals from one another. These datasets are mostly used for regression problems, as none of them are labeled. The datasets used are as follows:

1. The *bike_rental* (Fanaee-T & Gama, 2014) dataset contains of hourly data of bikes rentals in Washington, D.C., USA over two years. It contains data regarding weather, wind and the amount of casual and registered bike rentals. We used the total amount of bikes rented. Shape (17379, 15).

2. The *electricity* (ele) dataset contains 15 minute aggregated power consumption (KW) of 321 clients in Portugal over a year. The first client was used for prediction. Shape (26304, 321).

3. The *exchange* (exc) dataset contains the daily exchange rate between American Dollar and other currencies such as AUD, GBP, CAD, CHF, CNY, JPY, NZD and SGD in the years 1990 to 2016. Australian Dollar exchange rate was used for prediction. Shape (7588, 8).

4. The *solar* (sol) dataset contains solar power production records for 10 minute intervals in the year of 2006 from 137 photo-voltaic power station plants in Alabama, USA. The first plant was used for prediction. Shape (51560, 137).

5. The *traffic* (tra) dataset contains hourly data of road occupancy, recorded by different sensors in San Francisco freeways throughout the years 2015-2016. The first freeway was used for prediction. Shape (17544, 862).

6. The *turkey_power* (tur) dataset contains hourly electricity consumption (MWh) of Turkey in the years 2015 - 2020. Shape (63305, 16).

7. The *venezia_water* (ven) dataset contains hourly measurements from 1983 to 2015 of the water level in Venice (meters). Shape (289272, 1).

Some datasets were needed to be cleaned by replacing extreme or missing values with a default invalid value (in our case -1, occurred only in locations where -1 is indeed an invalid value). Furthermore, one-hot encoding of some features were added in *turkey_power* dataset. Additionally, all datasets were normalized to the range [0,1] as LSTM models suffer from the vanishing gradient problem and normalizing the data helps to avoid this problem. Window size was selected to be 48 time instances and prediction horizon of 12 was chosen.

### 4.6 PERFORMANCE EVALUATION

We use root mean squared error (RMSE) as our evaluating metric (see Equation 7), as we are predicting a feature, prediction horizon ahead. This metric captures how different is the predicted curve from the real measurement. Although exponential Partitioning can also be used for classification, we could not test this since the datasets we used are not labeled. Furthermore, RMSE itself is not sufficient to describe improvement as it depends on the scale of the data, we compare the decrease in the RMSE, compared to the baseline, as described in Equation 8.

$$RMSE = \sqrt{\frac{1}{n}\sum_{i=1}^{n}\left(t_i - p_i\right)^2} \qquad (7) \qquad Improv.\% = \frac{RMSE_{baseline} - RMSE_{result}}{RMSE_{baseline} \times 100} \qquad (8)$$

## 5    RESULTS

### 5.1    SETUP

For the experiments, we used $k$-fold validation, we divided each dataset into $k = 5$ parts, where for each iteration, we used one part as the test set and the other four as training. Finally, we averaged the results of all iterations as the final result.

The proposed model was implemented using the Keras (Chollet, 2015) library with TensorFlow backend (Abadi et al., 2016) using Python. Training the model was done on an Intel(R) Xeon(R) Platinum 8171M CPU in the Microsoft Azure cloud, where a single prediction takes about 5 milliseconds. The model used is a simple LSTM neural network with 10 neurons in order to facilitate fast experimentation. We also report an experiment with a few more complicated models to show that our results are not limited to simple models.

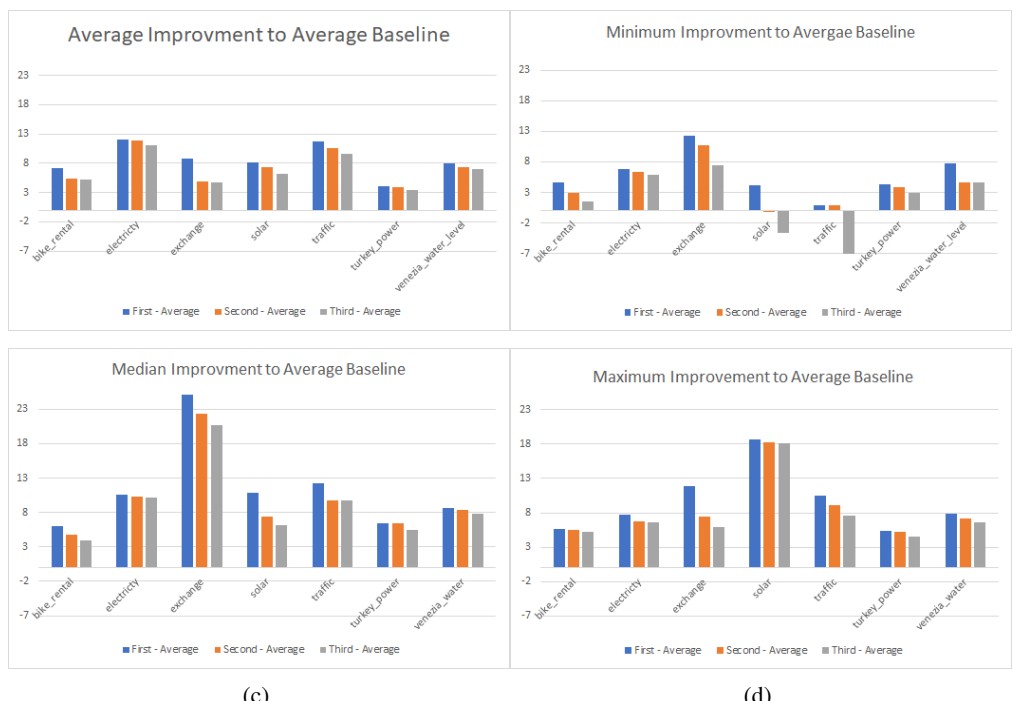

(c)                                                      (d)

Figure 4: Percentage of improvement compared to average uniform baseline in RMSE.

### 5.2    EXPONENTIAL PARTITIONING RESULTS.

We compared the results of the partitions in Table 1 with uniform baseline aggregated with average function and simple baseline (since all bins are 1, no aggregation is needed). Results of the comparisons with the uniform baseline can be seen in Figure 4a. For all the datasets the top 3 partitions improve the accuracy of the baseline by 3-12%. The top partition improvement was 4-12%. For the *electricity* and *traffic* datasets the top 3 partitions are all above 10% improvement.

When adding a different aggregation function to exponential Partitioning  improvement changes dramatically (see Figure 4b-4d, all the comparisons in the figures are made with the average aggregation function using a uniform partition). Median and maximum appears to perform best, where the median shows an improvement of 6-27%, and maximum 5-18%. Note that in many cases, the second and even the third best partition has similar improvement as the best partition. We will discuss this point later with additional data, but even at this point we can see that the improvement is not sensitive to some special partition. The minimum aggregation function does not show significant improvement in several datasets and in others datasets there are larger gaps between the top partitions performance. Thus, it is not an attractive candidate for an aggregation function.

It is interesting to isolate the improvement due to the aggregation function. For this purpose we bring Figure 5 and Table 2. Figure 5 compares the top three partitions of the three aggregation functions to

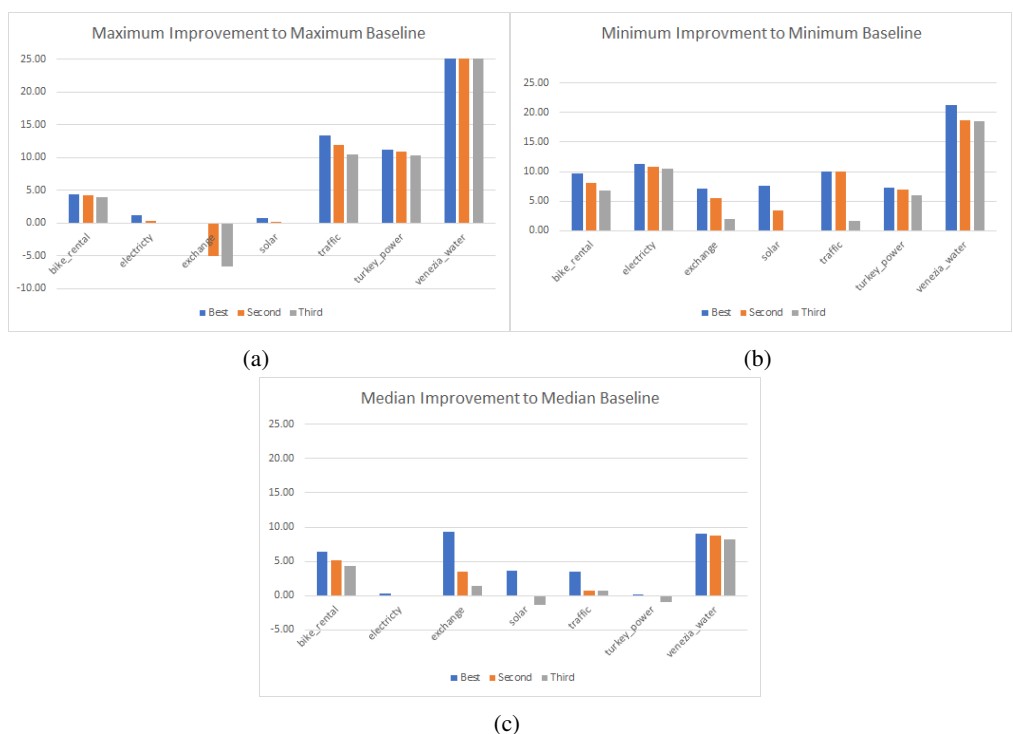

(a)                                                           (b)

(c)

Figure 5: Percentage of improvement compared to uniform baseline with the same aggregation function in RMSE. The bars for *venezia_water* for the maximum aggregation function were trimmed to obtain better visibility.

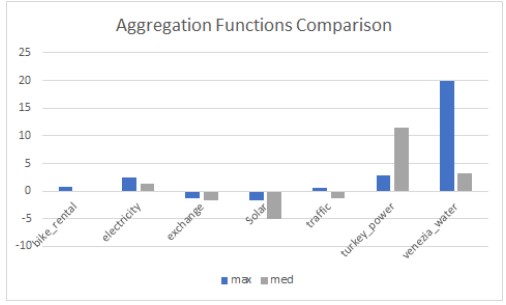

Figure 6: Comparison of different aggregation functions.

Figure 7: Comparison of 3 partitions used by different models

the baseline partition with same aggregation function. For the minimum function the improvement is always positive in the range of 7-21%. For the maximum we see even greater improvement in most cases, but for one dataset (*exchange*) there is a negative improvement. For median the differences are smaller. Figure 6 shows the improvement of the top partitions, with maximum and median aggregation functions, compared to the top partition with average aggregation.

Table 2 compares the performance of applying the three aggregation functions with the average on the uniform partition. The picture here is mixed: in two datasets the average function perform best, in three datasets median is the best and in two datasets maximum is the best, thus all the improvements are negative, in three datasets average is the second best, and only in one case all the other three aggregation functions are performing better than average.

Table 3 shows the best result achieved for each dataset. Aggregating with the median function is the best in 4 out of 7 of the datasets, while averaging is the best for only 2. As for the best partitions, all but one of them have the first 4 bin sizes equal to 1 or 2. There is a clear preference for highly skewed partitioning. Thus, when the best partitioning is exponential, it is with a very small $\varepsilon \in [0.1, 0.15]$.

| Function \Dataset | bike_rental | electricity | exchange | solar | traffic | turkey_power | venezia_water |
|---|---|---|---|---|---|---|---|
| MAX | 1.30 | -6.055 | 11.82 | 18.14 | -3.21 | -10.27 | -33.55 |
| MIN | -5.62 | -19.33 | 5.48 | -3.68 | -10.10 | -6.94 | -18.70 |
| MED | -0.42 | -1.82 | 19.59 | 7.37 | 9.05 | 3.04 | -1.70 |

Table 2: improvement of different aggregation functions over the average aggregation on uniform partitioning.

| Dataset | Function | Partition | $1 + \varepsilon$ | base (b) | Improv. from Uniform baseline(%) | Improv. from Simple baseline(%) |
|---|---|---|---|---|---|---|
| _bike_rental_ | AVG | {1, 1, 1, 1, 1, 2, 2, 39} | 1.15 | 1 | 7.15 | 8.40 |
| _electricity_ | AVG | {1, 1, 1, 1, 1, 2, 2, 39} | 1.15 | 1 | 12.13 | 11.53 |
| _exchange_ | MED | {4, 4, 4, 4, 4, 4, 4, 20} | – | – | 27.08 | 27.36 |
| _solar_ | MAX | {2, 2, 2, 2, 3, 3, 3, 31} | 1.10 | 2 | 18.72 | 42.09 |
| _traffic_ | MED | {1, 1, 1, 1, 1, 1, 1, 41} | – | – | 12.17 | 14.99 |
| _turkey_power_ | MED | {1, 1, 1, 1, 1, 1, 1, 41} | – | – | 6.43 | 21.66 |
| _venezia_water_ | MED | {1, 1, 1, 2, 4, 8, 15, 16} | – | – | 8.67 | 33.96 |

Table 3: Top results – generic.

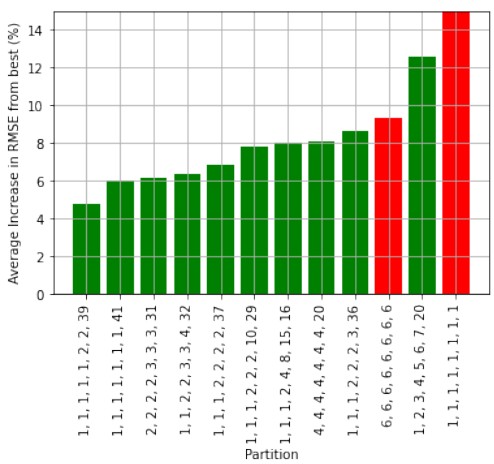

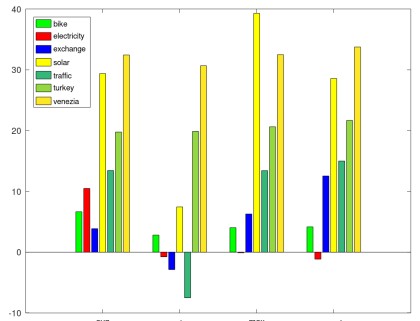

Figure 8: Percentage of increase in RMSE using different partitions (baselines are marked in red)

Figure 9: Performance improvement of the partition {1,1,1,1,1,1,1,41} over the partition {1,1,1,1,1,1,1,1}.

Showing only the best partition for each dataset is not sufficient when one wishes to understand how to select good candidate partitions.

As we mentioned before, in many cases the difference between the top three partitions is rather minimal. Thus, we tabulated in Figure 8 for each partition, the average increase of RMSE of all datasets, compared to the smallest RMSE, in the specific dataset. Interestingly {1,1,1,1,1,1,1,41} appears as one of the smallest RMSE, although it appears only twice in Table 3. The next top performing partitions are exponential, {1,1,1,2,4,8,15,16} and {1,1,1,1,1,2,2,39}. Then there are other skewed partitions, mostly exponential. The {4,4,4,4,4,4,4,20} partition appears once (we see it in Table 3). It still places significant weight (20) in the last bin, but it is not as skewed as the rest of the good partitions. In general, by experimenting with a few top partitions, one can easily find one that is close to the best.

It is already obvious that the partition {1,1,1,1,1,1,1,41} is performing rather well as can be seen in Table 3 and Figure 8. This partition differ from the simple baseline in the last bin, where the entire remaining history is aggregated. Figure 9 shows the performance improvement of the partition {1,1,1,1,1,1,1,41} over {1,1,1,1,1,1,1,1}. Despite the fact that all the remaining 40 historical samples are aggregated and added to the single last bin the improvement is significant for the majority of the cases. In only four cases the improvement is negative, three of them are when the minimum aggregation function is used, which was shown to be the worst function, by far.

All our results so far used a window size of 48 samples. As mentioned before, a larger window size does not mean higher accuracy when using uniform partitioning. Figure 5.2 shows the RMSE for the aggregation baseline partition and an exponential partitioning as a function of the window size for *solar*dataset. In this dataset, when the window size increases the RMSE of the aggregation baseline increases as well, but on the other hand, when using exponential partitioning, RMSE continues to decrease as the window size grows. Table 4 shows an example of the top 10 partitions for the experiment for the *solar* and a window size of 96. Interestingly, small $\varepsilon$ in the range [0.05,0.1] performs best, smaller than for window size 48.

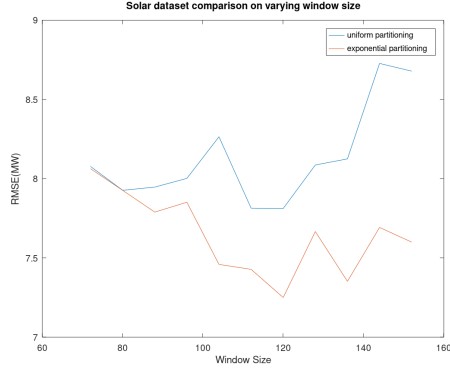

Figure 10: predicted RMSE using average aggregation function as a function of the window size for *solar*.

| Partition | $\varepsilon$ | base (b) |
|---|---|---|
| {5, 5, 5, 6, 6, 6, 6, 7, 7, 7, 7, 29} | 1.04 | 5 |
| {4, 4, 5, 5, 5, 6, 6, 7, 7, 8, 9, 30} | 1.08 | 4 |
| {5, 5, 6, 6, 7, 7, 8, 9, 9, 10, 11, 13} | 1.08 | 5 |
| {4, 4, 5, 5, 6, 6, 7, 7, 8, 9, 9, 26} | 1.09 | 4 |
| {4, 4, 5, 5, 6, 6, 7, 8, 9, 9, 10, 23} | 1.1 | 4 |
| {4, 4, 5, 5, 5, 6, 6, 6, 7, 7, 8, 33} | 1.07 | 4 |
| {4, 4, 4, 5, 5, 5, 5, 6, 6, 6, 7, 39} | 1.05 | 4 |
| {5, 5, 6, 6, 7, 7, 8, 8, 9, 9, 10, 16} | 1.07 | 5 |
| {7, 7, 7, 7, 8, 8, 8, 8, 8, 8, 9, 11} | 1.02 | 7 |
| {4, 5, 5, 6, 7, 8, 9, 11, 12, 14, 15} | 1.15 | 4 |

Table 4: Top 10 partitions for a window of size 96.

To show that our results are not limited to LSTM, we trained other models, GRU, stacked LSTM, RNN, LSTM Auto-encoder, GRU CNN, CNN and Bidirectional LSTM, on *solar*, with a window size of 96 that is aggregated using 3 different partitions: Simple baseline (i.e., all elements are 1), Uniform baseline (i.e., all elements are 8) and exponential Partitioning with $\epsilon = 0.09$ and $b = 4$. Figure 7 shows that all models, except RNN (which is not widely used anymore), achieve lower RMSE using exponential Partitioning , compared to the baselines.

## 6 CONCLUSION AND FUTURE WORK

In this paper, we showed that the common way of aggregating long time series to feed LSTM networks, by partitioning the data to equal size intervals (uniform partition) and representing each interval with the average value of the samples in it, is far from optimal. We suggested non-uniform partitioning and the use of other aggregation functions to obtain significant improvement in prediction.

Our finding show that while the best data partition depends on the dataset, there are a few partitions that tend to perform best or close to best for all datasets. Thus, to obtain good prediction one needs to experiment only with a few candidate partitions, that are based on exponential partitioning with small $\varepsilon$ values. In our experiment we found that almost every partitioning that use smaller bin sizes for the near history and larger ones for the far history, and maintain a non-decreasing bin size order is performing better than uniform partitioning. The cases where the uniform partitioning was performing relatively well, occurred when the differences where very small. We believe that future work should better understand how to select the best exponent for the partition.

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
