# OpenReview forum: "A Study of Aggregation of Long Time-series Input for LSTM Neural Networks"
_ICLR.cc/2022/Conference — ICLR 2022 Submitted_

### Official Review · Reviewer_nfPs · 2021-10-25

**Correctness:** 2
**Technical Novelty And Significance:** 2
**Empirical Novelty And Significance:** 1
**Recommendation:** 3
**Confidence:** 4

**Main Review:**

Strengths

-  Use seven publicly available datasets to evaluate the proposed partitioning procedure.
-  Source code is available to reproduce the results.

Major Weaknesses

- In the abstract, authors phrase, “Long-short term memory (LSTM) models are the state-of-the-art for time-series forecasting”, which is a very strong claim to highlight given the recent success of other machine learning architectures in the time series forecasting space. For example, the winning solution of the M5 forecasting competition was based on the LightGBM model [1], whereas the best performing model of the M4 forecasting competition was a hybrid architecture of Recurrent Neural Networks and Holt-winters model (EN-RNN) [2]. The results of these competitions are often used as a yardstick to measure the quality of newly proposed forecasting methods.  The common attribute of these best performing models is that they are trained across a collection of time series and share the same set of parameters across all-time series (Global Forecasting Models). Therefore, a more appropriate statement would be that the current state-of-the-art in time series forecasting is moving from univariate forecasting models to global forecasting models. Refer [3] for more discussions around univariate models vs global models. I agree that the LSTM architecture can be a strong architecture for time series forecasting (preferably in a GFM setting, read [4]), but referring LSTM as “the state-of-the-art” can be misleading here.

-  Therefore, authors have failed to mention key related work applicable to GFMs, which also raises the question about the training mechanism used to train the proposed models in this study (LSTM and other RNN variants such as GRU). Are they univariately or globally trained ?

- Even though authors use seven benchmark datasets (which is commendable) to evaluate the proposed exponential partitioning procedure, they have not compared against current state-of-the-art techniques in the time series forecasting space, or haven’t used the proposed partitioning technique to train modern forecasting architectures such as DeepAR, NBEATS ES-RNN, Transformer, WaveNet etc. Ideally, authors could have used this partitioning procedure to train these architectures with longer time series sequences and attest their respective performance. This would help authors to make strong claims about the effectiveness of this method with current state of the art.

- Authors should use error measures that are commonly used in the forecasting literature (MASE, sMAPE). RMSE can be highly sensitive to outliers. Please refer to [1], where the limitations and drawbacks of the MAPE error measure has been extensively discussed [5]. Moreover, authors also do not show whether the results are statistically significant.

-  Authors also do not discuss the hyper-parameter optimisation method used to optimise the various hyper-parameters of the models (LSTM, CNN)

Minor comments

- Please proofread the paper again (grammar, language use, etc)
- Make sure to introduce the acronym first, before being used later in the text (e.g CNN it not defined as Convolutional Neural Networks in the text)


[1] Makridakis, S., & Spiliotis, E. (2021). The M5 competition and the future of human expertise in forecasting. Foresight: The International Journal of Applied Forecasting, 60, 33–37.

[2] Smyl, S. (2020). A hybrid method of exponential smoothing and recurrent neural networks for time series forecasting. International Journal of Forecasting, 36, 75-85

[3] T Januschowski, J. Gasthaus, Y. Wang, D. Salinas, V. Flunkert, M. Bohlke-Schneider, and L. Callot. “Criteria for classifyingforecasting methods”. In:Int. J. Forecast.36.1 (2020), pp. 167–177

[4] Hewamalage, H., Bergmeir, C., & Bandara, K. (2021). Recurrent neural networks for time series forecasting: Current status and future directions. International Journal of Forecasting 37, 388–427

[5] Hyndman, R.J. and Koehler, A.B. 2006. Another look at measures of forecast accuracy. International journal of forecasting. (2006)





**Summary Of The Paper:**

In this study, authors propose an exponential partitioning procedure to minimise the vanishing gradient and exploding gradient problems that can occur when forecasting time series with long time series. Using 7 publicly available time series datasets, authors show that the proposed exponential partitioning procedure is able to outperform the traditional uniform partitioning procedures.

**Summary Of The Review:**

Even though the motivation of this work is clear, I have multiple concerns about the scientific rigour of the experimental setup (see the review section). Based on the current status of the manuscript, I am recommending to reject this paper.

---

### Official Review · Reviewer_tJW4 · 2021-10-26

**Correctness:** 3
**Technical Novelty And Significance:** 2
**Empirical Novelty And Significance:** 2
**Recommendation:** 3
**Confidence:** 5

**Main Review:**

## Pros
- I think the idea is an interesting one with respect to shorter aggregation intervals, its relatively simple but I've not seen it considered elsewhere.
- Performance over the proposed baseline is convincing over the datasets considered.

## Cons
I apologies that my main con is that I do not think this question/methodology is of sufficient interest to warrant acceptance to this conference. The work is okay and, assuming the idea of exponential change of window size away from the prediction time has not been considered elsewhere, should be published somewhere. However, I do not think there is sufficient novelty or new ground being broken in this paper to warrant acceptance.
- It's bold to still call LSTMs state-of-the-art for time series forecasting, check any meta-review these days and there tend to be many better performers.
- Tables and graphics are non-consistent and not nice on the eyes. Check some papers that get accepted to ICLR/NeurIPS and emulate the table styles, Figures 8 and 9 are the worst graphic offenders since they have a complete change of color scheme from the previous ones that is not pleasant viewing.

**Summary Of The Paper:**

The authors consider the problem of learning from long time series where the model (e.g. LSTM) cannot accept the full time series directly (due to memory/time constraints) and instead must accept some aggregated short form of the time series. The standard method for this is aggregation over equal-sized bins. In this paper, the authors introduce an exponential partitioning, whereby the time series is split into bins but the bin sizes increase as we move further from the current time. The rationale is that the time segments closer to the current time contain more value for future predictions and thus would benefit from aggregation over shorter time scales.

The authors test the approach for a few different aggregation functions (mean/median/min/max) and show good improvement of the proposed method compared with the simple baseline.

**Summary Of The Review:**

Okay but its a way from being accepted to ICLR or something else top tier.

---

### Official Review · Reviewer_Fmnr · 2021-10-28

**Correctness:** 3
**Technical Novelty And Significance:** 1
**Empirical Novelty And Significance:** 2
**Recommendation:** 3
**Confidence:** 3

**Main Review:**

__Pros.__

1. The paper studies the benefits of exponential aggregations scheme which are often used in practical time-series forecasting applications. This is because due the the sheer volume of data in industrial time-series datasets, it might make sense to store older data in more coarsely aggregated form versus more recent data. This technique is also used in practical seq-2-seq models to ensure that the encoder can attend to a larger history. Even though it is used in practice, I have not seen a systematic study in the literature. So an empirical study would be valuable if done properly.

2. The paper does a decent job of describing the details of each of the experiments performed, so it is should be fairly reproducible. I do need some more clarifications, that are mentioned later in the review.


__Cons.__

1. The paper does not discuss or compare with well known advances in architectures that are specifically designed to attend to longer histroy length. For example causal convolution architectures with exponentially increasing architecture (wavenet architectures) have been used successfully for multi-variate time-series forecasting. Some references are:

--- Anastasia Borovykh, Sander Bohte, and Cornelis W Oosterlee. Conditional time series forecasting with
convolutional neural networks.

--- Sen, Rajat, Hsiang-Fu Yu, and Inderjit Dhillon. "Think globally, act locally: a deep neural network approach to high-dimensional time series forecasting." Proceedings of the 33rd International Conference on Neural Information Processing Systems. 2019.

These architectures can attend to history length exponentially increasing in depth and can have a similar effect as exponential aggregation in LSTM. It would be good to cite and compare with these architectures for the sake of completeness.

2. In the introduction the paper states that : _The mentioned systems can be described using a single variable over time, namely univariate time series, or several variables altogether as a multivariate time series. In both cases, one variable is
marked as the important one whom we’d like to predict._ This is not true, as in many retail applications we are interested in forecasting the demand of all the items not just one. Indeed the paper considers multivariate forecasting datasets but only predicts one time-series. I think this limits the impact of the study. It would we great if the paper can use the same well-established multi-variate forecasting benchmarks as used in https://arxiv.org/pdf/2103.07719.pdf that is they should report metrics on multi-variate forecasting tasks.

3.  I do not think Section 2 adds any value to the paper and is not directly related to the problem being studied.

4. The whole point of this paper is to display the benefits of longer history but in all the experiments a maximum history length of 48 is considered which is not long and a LSTM architecture might be able to handle it. I would be interested to know the performance of just feeding each point to the network i.e in Table 1 convention {1, 1, ....., 1} (48 ones). Also, what are the basis of choosing the window schemes in rows 2-6 in Table 1?

5. It would be good to see the performance of the best window and aggregation scheme with a regular LSTM on one the standard multivariate forecasting tasks for instance the ones in https://arxiv.org/pdf/2103.07719.pdf, in order to judge whether just a better aggregation scheme is enough to boost the performance of the LSTM architecture into top-k in terms of SOTA performance.

6. In most experiments the task is to predict one our of many time-series in the datasets. In this case are the values of the other time-series used as covariates? If so are they fed into the input layer of the LSTM directly? I could not find a clarification of this in the paper, it would be good if you can point me to it.

7. There might be some additional references:
--- https://arxiv.org/pdf/2012.08041.pdf -> learnt non-uniform temporal aggregation scheme for action recognition tasks
--- http://proceedings.mlr.press/v54/bhowmik17a.html --> frequency domain learning of non-uniformly aggregated data


**Summary Of The Paper:**

The paper studies different windowing schemes and pooling schemes for feeding data into an LSTM encoder for time-series forecasting. The windowing schemes considered are different uniform divisions of the history and exponential window sizes where data further back in the past is aggregated over larger windows. The latter enables the network to get fine grained information from more recent data while also attending to larger history length. The paper also compares different aggregation or pooling functions like max, min, mean and median. Overall there seems to be some value in exponential windowing and median aggregation scheme based on the experiments on 7 datasets, where the task is to predict the values of only one univariate time-series.

**Summary Of The Review:**

Overall there is value in studying the problem. However, as evident from the Cons above the paper falls short of doing a thorough empirical study. There are multiple problems like not studying longer history length problems, not comparing on well-known multi-variate forecasting benchmark tasks, and not comparing with a simple baseline that feeds all the granular data (this can be done for history length 48).

---

### Official Review · Reviewer_2ThG · 2021-10-30

**Correctness:** 2
**Technical Novelty And Significance:** 1
**Empirical Novelty And Significance:** 3
**Recommendation:** 3
**Confidence:** 4

**Main Review:**

The main idea in the paper -- of using non-uniform aggregation of inputs to LSTM is reasonable, and I am not surprised that it seems to show promise. I have seen a variety of related published ideas about multi-scale aggregation (authors cite a few), but I can't think of a paper that proposed this specific simple idea. Likewise, the idea of using non-linear aggregation functions over bins is also reasonable. The paper is purely experimental, and does show evidence that some form of non-uniform aggregation would be a good idea to try when dealing with long time-series. While the proposal is simple, it could have practical significance.

Unfortunately, the paper has various issues, so I can't recommend its publication:
1) There are various issues with the experimental setup:
a) There are 7 datasets and 13 different choices of partitions. Many studies report the best among 13 partitions for each dataset -- so it's not surprising that some partition other than the baseline will win the contest for each specific dataset.  The choice of the specific 13 partitions seems fairly arbitrary.  Some are "exponential-like", some are uniform with different bin-widths, some are linear,  many are basically uniform with short memory except for the last large bin.
b) There is some evidence that front-loaded partitions e.g. (1,1,1,1,..., 41) seem to be helpful for several datasets.  However, I think this is basically an LSTM with a window of size 8, which also has an additional longer-term mean feature. It's not really an 'exponential' partition.
c) There is no attempt at statistical significance.
d) For datasets which have many time-series (e.g. ele, exc, sol, e.t.c.) the authors only pick 1 timeseries for analysis, and drop the others.
e) The paper uses 5-fold cross-validation on time-series examples -- no details are provided -- but interpreting it naively it seems to use past and future data in training, and test data is in the middle. This isn't a great choice for time-series.

2) The example in section 2 is not at all convincing (and unclear) -- two specific functions are selected (log and sin), and the authors argue that picking two points in a non-uniform way works better.  For any partition one can select a function for which that partition is optimal. There's no theory.  Notation is unclear.  E.g. "k" is undefined.  I assume it's the split point.

3) There is only a very minimal attempt to generalize / interpret the findings. What happens when the input window is not length 48, and when you're not predicting 12 steps ahead? When is max/min/median helpful and why compared to the average. Why is max helpful but min doesn't seem to be?  If you negate your input time-series -- then original mins become maxes -- wouldn't it now make min be a good choice for aggregation now?  What is the typical limit of how many time-steps an LSTM can handle, what does it depend on?  How can you concisely parameterize the choice of interesting partitions to explore? (E.g. the 13 in the paper).  And how to apriori know which partition may be helpful for what kind of time-series (maybe based on auto-correlation / spectral properties).

4) There is no mention / comparison to the rich transformer literature (or other attention models), including the various models addressing long-memory -- e.g. longformers, reformers, e.t.c.

5) Captions in the figures are hard to understand, what does "Average improvement to average baseline" mean?  Experimental setup is also not entirely clear, and requires a few passes to understand what's being reported.

6) Various minor grammar issues -- too many to mention here.

**Summary Of The Paper:**

This paper looks at how to aggregate time-series inputs to LSTM models, and recommends to use non-uniform aggregation, with small recent bins and larger older bins. Furthermore, instead of simple averaging of time-samples within the bin, it also found value by using non-linear aggregations like median, max and min.  The observations are mostly experimental using a collection of 7 time-series datasets.

**Summary Of The Review:**

This is an experimental paper that argues for using non-uniform (exponential) and non-linear (max and median vs avg) aggregation of long time-series inputs when used with LSTMs.  The ideas are simple, but potentially could be quite effective for some applications. Unfortunately, experimental results are flawed, and only partially convincing.

---

### Decision · Program_Chairs · 2022-01-20

**Decision:**

Reject

**Comment:**

This paper addresses unique windowing schemes for the input of an LSTM model for time-series forecasting, in particular an exponential partitioning, where bin sizes increase as moving further from the current time point. Although the basic idea is interesting and motivating and experimental results are strong; as reviewers pointed out, technical significance and novelty are limited because of lack of theoretical or conceptual justification and motivation the proposed approach.  The authors’ claim is primarily based on experiments results. Other critical issues include the lack of comparison with recent advances in specifically designed to attend to longer history length or the discussion of modern approaches. other issues include presentation (e.g., grammatical errors) and the use of acronyms before introducing them.